# Enhanced Antioxidant Effects of the Anti-Inflammatory Compound Probucol When Released from Mesoporous Silica Particles

**DOI:** 10.3390/pharmaceutics14030502

**Published:** 2022-02-24

**Authors:** Michael Lau, Benjamin Sealy, Valery Combes, Marco Morsch, Alfonso E. Garcia-Bennett

**Affiliations:** 1School of Natural Sciences, Macquarie University, Sydney, NSW 2109, Australia; michael.lau@mq.edu.au; 2Malaria and Microvesicles Research Group, School of Life Science, Faculty of Science, University of Technology Sydney, Ultimo, Sydney, NSW 2007, Australia; benjamin.sealy@uts.edu.au (B.S.); valery.combes@uts.edu.au (V.C.); 3Centre for Motor Neuron Disease Research, Department of Biomedical Sciences, Faculty of Medicine, Health and Human Sciences, Macquarie University, Sydney, NSW 2109, Australia; marco.morsch@mq.edu.au; 4Australian Research Council Industrial Transformation Training Centre for Facilitated Advancement of Australia’s Bioactives (FAAB), Macquarie University, Sydney, NSW 2109, Australia

**Keywords:** probucol, solubility, blood brain barrier, inflammation, oxidative stress, neuroinflammation, mesoporous silica particles

## Abstract

Brain endothelial cells mediate the function and integrity of the blood brain barrier (BBB) by restricting its permeability and exposure to potential toxins. However, these cells are highly susceptible to cellular damage caused by oxidative stress and inflammation. Consequent disruption to the integrity of the BBB can lead to the pathogenesis of neurodegenerative diseases. Drug compounds with antioxidant and/or anti-inflammatory properties therefore have the potential to preserve the structure and function of the BBB. In this work, we demonstrate the enhanced antioxidative effects of the compound probucol when loaded within mesoporous silica particles (MSP) in vitro and in vivo zebrafish models. The dissolution kinetics were significantly enhanced when released from MSPs. An increased reduction in lipopolysaccharide (LPS)-induced reactive oxygen species (ROS), cyclooxygenase (COX) enzyme activity and prostaglandin E_2_ production was measured in human brain endothelial cells treated with probucol-loaded MSPs. Furthermore, the LPS-induced permeability across an endothelial cell monolayer by paracellular and transcytotic mechanisms was also reduced at lower concentrations compared to the antioxidant ascorbic acid. Zebrafish pre-treated with probucol-loaded MSPs reduced hydrogen peroxide-induced ROS to control levels after 24-h incubation, at significantly lower concentrations than ascorbic acid. We provide compelling evidence that the encapsulation of antioxidant and anti-inflammatory compounds within MSPs can enhance their release, enhance their antioxidant effects properties, and open new avenues for the accelerated suppression of neuroinflammation.

## 1. Introduction

Endothelial cells play a key role in the development and function of blood and lymph systems. They form the physical barrier between blood and the brain interstitial fluid, restricting the passage of molecules that potentially harm the brain, while facilitating the movement of essential nutrients across the blood brain barrier (BBB) [1]. These endothelial cells are among the first cells to be exposed to pro-inflammatory stimuli such as bacterial or viral pathogens [2]. These stimuli can elicit a strong immunological response in the host cell, increasing the production of reactive oxygen species (ROS) and inflammatory mediators that act as messengers that either activate the host defense mechanisms or recruit professional immune cells to contain and eliminate the pro-inflammatory stimuli. Antioxidants serve as the first line of response after such inflammatory stressors. They maintain ROS within normal physiological concentrations and contribute to essential cellular functions such as sustaining the integrity of the BBB, fostering immune responses and operating as a control mechanism between cell viability and apoptosis [3].

Chronic exposure to high doses of pro-inflammatory stimuli can induce cellular oxidative stress and inflammation. Cyclooxygenases (COX) are pro-oxidant enzymes that produce the superoxide anion O_2_^−^, the main oxidant from the metabolism of arachidonic acid and its conversion to prostaglandins [4,5]. The superoxide anion is rapidly converted to more reactive secondary ROS including hydrogen peroxide, hydroxyl and peroxyl radicals [6]. Furthermore, the superoxide anion can react with nitric oxide to generate peroxynitrite, a highly reactive oxidant [6]. Inhibition of COX-2, the primary cyclooxygenase involved in inflammation, prevents the production of prostaglandins, thereby reducing inflammation. Other COX-2-independent mechanisms are also known to be relevant in the regulation of pro-inflammatory molecules in neurodegenerative processes [7].

In the context of the BBB, oxidative stress leads to the production and release of high concentrations of ROS that affect its integrity due to impairment of tight junction proteins and cell apoptosis via oxidation of biological membranes, DNA, proteins and lipids [8]. The increase in BBB permeability is a key event in the pathogenesis of neuroinflammatory diseases in Parkinson’s, Alzheimer’s and multiple sclerosis patients. Thus, it is crucial to investigate potential pharmaceutical compounds with improved antioxidant and anti-inflammatory properties that can maintain the integrity of the BBB [9]. One approach is the use of drug compounds that reduce COX enzyme activity. The formation of prostaglandin E_2_ (PGE_2_) by the COX-dependent pathway in brain endothelial cells has been found to mediate the inflammatory response in immunologically challenged mice following a peripheral injection of lipopolysaccharide (LPS). As a bacterial endotoxin, LPS, binds to toll-like receptor 4 (TLR4) on the surface of endothelial cells, initiating the production of inflammatory mediators such as tumor necrosis factor (TNF-α) and ROS [10,11,12,13]. PGE_2_ also disrupts the integrity of the BBB by increasing vascular permeability and edema [14,15]. Importantly, the reduction in COX-mediated production of PGE_2_ has been shown to attenuate the inflammatory response [16,17,18].

The diphenolic compound probucol (PB) has been shown to attenuate increased activity of the COX-2 enzyme via the NF-κB pathway. This leads to reduced expression of pro-inflammatory markers [19,20] and decreased disruption of the BBB integrity in mice [21,22]. Additionally, PB is a potent free radical scavenger and is oxidized by free radicals in the formation of stable metabolites, protecting biological molecules and cellular membranes against oxidative damage [23,24]. However, PB is a highly lipophilic and poorly soluble compound, requiring a long pre-incubation time of up to 8 h at a high dosage of 50 μM in order for a sufficient amount of drug to be transported to endothelial cells [22]. Lipophilic drug carriers are required to facilitate oral absorption of PB, limiting its current application in the treatment of metabolic and vascular diseases, as well as its potential in combating neuroinflammatory diseases [25,26,27,28,29].

Mesoporous silica particles with defined pore structures [30,31,32], tunable pore sizes and high surface area are known to enhance the solubility of a range of poorly soluble antioxidant and anti-inflammatory drugs including PB [33,34,35,36,37,38]. Cui et al. [39] have demonstrated an enhanced delivery of the antioxidant resveratrol (RSV) loaded at very low levels (1.6 wt.%), using a combination of polylactic acid and low-density lipoprotein receptors coated on mesoporous particles (200 nm average particle diameter, 4 nm pore size) using an in vitro BBB endothelial/microglia cell model. The released RSV decreased oxidative stress of LPS-activated microglia cells, specifically reducing levels of the inducible nitric oxide synthase enzyme [40]. The aqueous solubility of RSV is 0.03 mg/mL, whilst that of PB is 4.18 × 10^−5^ mg/mL. Improving the solubility of these compounds enhances their pharmacological properties and can expand their therapeutic indications.

In this work, we measure levels of cellular oxidative stress and inflammation caused by LPS and their attenuation using PB when released from MSPs, using an in vitro brain endothelial cell model. The MSP known as AMS-6 (Anionic Mesoporous Silica-6) is chosen because of its reproducible, scalable and easy synthesis using the amino acid-derived anionic surfactant *N*-lauryl-alanine. Mesoporous AMS-6 possesses a spherical morphology and 3-dimensionally connected cylindrical pores, which afford a rapid release and dissolution profile of poorly soluble compounds [41,42,43].

We confirm that PB can significantly reduce oxidative stress burdens when released from AMS-6 as compared to free PB and ascorbic acid in an in vivo model of oxidative stress in zebrafish. Zebrafish display a high genetic homology (and conservation) in organ structure to humans, making it a relevant biological model in the study of the pathological mechanisms that are relevant to human diseases [44,45,46,47,48,49,50]. Similar to humans, ROS produced by zebrafish serve as second messengers in normal physiological function and also in the immune response following exposure to potentially harmful stimuli including environmental pollutants, chemicals and drug compounds [51,52,53]. Oxidative stress is induced in zebrafish using hydrogen peroxide (H_2_O_2_) as the pro-oxidant. The compound 2′,7′-dichlorofluorescin (DCFDA) is used as a proxy measure of intracellular oxidant concentration [15,54,55,56].

## 2. Materials and Methods

### 2.1. Material

Tetraethyl orthosilicate (TEOS) and 3-aminopropyl triethoxysilane (APTES) were purchased from Sigma Aldrich (Sydney, Australia) as reagent grade, and used without further purification. Throughout the experiments Milli-Q water was used. Probucol was purchased from Sigma Aldrich (Sydney, Australia) and used without further purification.

### 2.2. Synthesis of AMS-6

The mesoporous silica AMS-6 was synthesized following a protocol reported previously [57]. Briefly, the amino acid surfactant *N*-lauroyl-L-Alanine (C_12_Ala) was used together with APTES as the co-structure directing agent (CSDA). TEOS was used as the silica source. A solution of C_12_Ala (0.10 g) in Milli-Q water (20 g) was prepared and heated at 80 °C in a closed lid PVC bottle for 24 h under constant stirring to homogenize. After 24 h, APTES (0.10 g) was added to the solution, followed by TEOS (0.51 g), and the solution stirred for a further 24 h at 80 °C. After 24 h, the synthesis gel was stored at room temperature (RT) for 12 h. The silica product was filtered and dried at RT resulting in the as-synthesized AMS-6 (AS-AMS-6). The AS-AMS-6 sample was calcined in a furnace at 550 °C for 3 h with an initial heating rate of 0.8 °C/min was conducted to remove the surfactant. The product of the calcination is termed CAL-AMS-6.

Amine-functionalized AMS-6 (NH_2_-AMS-6) was prepared from AS-AMS-6 material by refluxing the solid (1 g) for 24 h at 85 °C in a 500 mL of an ethanol/HCl (37%) mixture, as previously reported [58]. Extraction of the as synthesized material resulted in the removal of surfactant, revealing the covalently bound propyl amine groups on the surface of silica. This allows further coupling with the fluorochrome, fluorescein isothiocyanate (FITC). The mixture was filtered, and the material was washed with ethanol and left to dry at room temperature overnight.

Preparation of FITC-AMS-6 MSPs was performed by the reaction between NH_2_-AMS-6 with the fluorochrome under mild alkaline conditions to produce the iminothioester bond [59]. Briefly, 1 g of NH_2_-AMS-6 was refluxed in 100 mL ethanol containing the desired amount of fluorochrome, and the pH of the mixture was raised to 11 by the addition of NaOH pellets. The mixture was allowed to reflux at 85 °C for a period of 6 h. The mixture was filtered and washed with ethanol to dissolve any unattached FITC. The sample was dried at RT overnight.

### 2.3. Scanning Electron Microscopy

Images were obtained using a JSM-7401F scanning electron microscope (JEOL Ltd., Tokyo, Japan) operating at 1–2 kV with no gold coating, using gentle beam mode at magnifications 5–50,000.

### 2.4. Dynamic Light Scattering (DLS)

Experiments were performed with a Zetasizer ZS (Malvern Instruments, Worcestershire, UK), with a 173° detector angle, at 25 °C with a He-Ne laser (633 nm, 4 mW output power) as a light source. AMS-6 particles (1 mg/mL) were mixed with 1 mL filtered Milli-Q water and filled into disposable folded capillary cell (DTS1070) (Malvern Instruments).

### 2.5. Powder X-ray Diffraction (XRD)

Powder XRD was performed on free drug and loaded silica samples to determine the presence of the crystalline drug (Bruker D8 Discover XRD) using CuKα radiation (λ = 1.5418 Å). The diffraction patterns were recorded between 0.5 to 70° (2θ) for drug-loaded samples and 1 to 8° (2θ) for unloaded calcined mesoporous samples.

### 2.6. Fourier Transform Infrared Spectroscopy

Fourier Transform Infrared Spectroscopy (FTIR) spectra of samples were obtained using a Thermo Scientific Nicolet iS5 FT-IR Spectrometer with iD5 ATR accessory; in transmittance, from 4000 cm^−1^ to 400 cm^−1^ was averaged over 32 scans for each curve. All samples were measured without dilution.

### 2.7. Nitrogen Adsorption/Desorption Isotherm

Nitrogen adsorption/desorption isotherms were measured at liquid nitrogen temperature (−196 °C) using a Micromeritics TriStar II volumetric adsorption analyzer (Micromeritics Instrument Corporation, GA, USA) for calcined and drug-loaded mesoporous samples. Before the measurements, the samples were de-gassed for 3 h at 60 °C. The surface area of the samples were calculated by using the Brunauer–Emmett–Teller equation in the relative pressure (P/P_o_) range of 0.05 and 0.3 [60]. The total pore volume was calculated from the amount of gas adsorbed at P/Po = 0.95. The pore size distribution curves were derived using the density functional theory assuming a cylindrical pore model for all samples [61].

### 2.8. Drug Loading

Pharmaceutical drugs were loaded into mesoporous silica via a wetness impregnation method. Calcined mesoporous silica was added to the ethanolic solution containing the dissolved compound and sonicated for ~10 min. The solvent was removed by rotary evaporation at 40 °C, with 150 rcf rotation at a pressure of 66 Pa. Samples were left to dry and stored in a desiccator until further use. Samples are named according to the drug-loading weight percentage (wt.%).

### 2.9. Thermogravimetric Analysis (TGA)

A thermogravimetric analysis instrument (TA instruments, TGA-2050, New Castle, DE, USA) was used to determine the drug loading amount in the mesoporous silica after loading. Analysis was conducted at a heating rate of 20 °C min^−1^ from 20 to 800 °C. The sample weights varied from 5 mg to 10 mg. The derivative weight loss calculation was performed using TA instruments software (TA instruments, Universal analysis 2000, version 3.0 G).

### 2.10. Differential Scanning Calorimetry (DSC)

A differential scanning calorimetry instrument (TA instruments, DSC-2010, New Castle, DE, USA) was used to determine the crystallinity of the sample at a heating rate of 10 °C min^−1^ from 20 to 350 °C. The sample weights varied from 5 mg to 10 mg. Analysis was performed using TA instruments software (TA instruments, Universal analysis 2000, version 3.0 G).

### 2.11. Drug Dissolution and Release

Drug release studies were conducted in simulated intestinal fluid (SIF) containing 0.25% (*w*/*v*) CTAB as a wetting agent prepared by dissolving NaOH (0.896 g, 1.792 g) and KH_2_PO_4_ (6.805 g, 13.61 g) in purified water (1 L) to yield a solution with a pH of 6.8. Size 1 gelatin capsules (ProSciTech, Batch: RL042, Queensland, Australia) were used to encapsulate both the pure drug and drug-loaded mesoporous silica samples. Drug release was assessed under sink conditions (900 mL SIF, pH 6.8) using a UV/Vis spectrometer (Agilent, Cary 60 UV-Vis, Sydney, Australia). The release was carried out in a dissolution bath (Agilent, 708-DS, Sydney, Australia) at a stirring rate of 50 rpm at 37 °C, and data were collected approximately every 5 min for 24 h.

### 2.12. Cells and Cell Culture

Human brain microvascular endothelial cells (cells kindly received from the group of Dr. Bingyang Shi, Macquarie Medical School, Sydney, Australia) were cultured in supplemented EBM-2 medium (Lonza, Basel, Switzerland; Cat. No. CC-3156) and grown in either 24- or 96-well plates pre-coated with rat-tail collagen type I (BD Biosciences, Franklin Lakes, NJ, USA) inside an incubator at 37 °C, 5% CO_2_–humidified atmosphere. All experiments were conducted under Macquarie University Biosafety Ethics Application (reference number 520221111136082). Cells of passage between 6 and 8 were used for experiments. When cells reached approximately ~90% confluency, cells were incubated with 1 μg/mL with or without PB or AMS-6PB for the duration of the experimental time points. The following assay kits were used to detect differences in the production of reactive species, cell viability and pro-inflammatory markers of the biological samples. A microplate reader (FLUOstar OPTIMA, BMG LABTECH, Mornington, VIC, Australia) was used for the analysis of biological samples using the assay kits.

### 2.13. Oxidative Stress and Cell Viability Assays

The Muse^®^ oxidative stress kit (abacus dx, Meadowbrook, QLD, Australia) provides a quantitative measurement of the percentage of cells that are under oxidative stress (%ROS+) from cells that are not under oxidative stress (%ROS−) based on the fluorescent detection of intracellular superoxide radicals. Each sample was prepared and analyzed according to the protocol provided by the manufacturer. Data analysis was performed on the MUSE cell analyzer^®^.

The Muse™ count and viability (abacus dx, Meadowbrook, QLD, Australia) reagent contain a DNA-binding dye that stains the nucleus of dead and dying cells while live cells are unstained by the reagent, allowing for the quantitative measurement of cells that are alive (viable) versus cells that are non-viable. Samples were prepared and analyzed following the protocols provided by the manufacturer.

### 2.14. DCFDA Cellular Reactive Oxygen Stress Measurement

The fluorescent probe 2′,7′-Dichlorodihydrofluorescein diacetate (ab113851, DCFDA Cellular ROS Detection Assay Kit, Abcam, Melbourne, Australia) was used for the detection of intracellular ROS including hydrogen peroxide, the peroxyl radical and peroxynitrite [62]. DCFDA is deacetylated by cellular esterase to the non-fluorescent 2′,7′-dichlorodihydrofluorescein (H_2_DCF) product that is unable to diffuse out of the cells. The presence of intracellular ROS oxidizes H_2_DCF to the fluorescent 2′-7′dichlorofluorescein (DCF), which is detected using the microplate reader at 485 nm excitation and 535 nm emission wavelengths [62]. Methods for sample preparation and data analysis followed the protocol suggested by the supplier.

### 2.15. Mitochondria Hydroxyl Assay

The mitochondrial hydroxyl radical detection assay kit (ab219921, Abcam, Melbourne, Australia) was used to detect intracellular hydroxyl radical using the OH580 probe, which is cell permeable and reacts with the hydroxyl radical to generate a red fluorescence detected at 540 nm excitation and 590 nm emission wavelengths.

### 2.16. Nitric Oxide Assay

The nitric oxide assay kit (ab65328, abcam, Melbourne, Australia) was used to measure the concentration of nitric oxide in the endothelial cells under the experimental conditions. The method for sample preparation and analyses followed the supplier’s recommended protocol.

### 2.17. Peroxynitrite Assay

The peroxynitrite assay kit (ab233469, abcam, Melbourne, Australia) contains the peroxynitrite sensor green, which specifically reacts with peroxynitrite in media to form a bright green fluorescent product measured at Ex/Em wavelengths of 490/530 nm. The protocol as recommended by the supplier was followed.

### 2.18. COX Enzyme Activity

Total COX enzyme activity was determined by the COX activity assay kit (ab204699, abcam, Melbourne, Australia). Methods for sample preparation and analysis followed the protocol as recommended by the supplier. The microplate reader was used to measure fluorescence at excitation/emission wavelengths of 535/587 nm in a kinetic mode at room temperature for 30 min.

### 2.19. PGE_2_ Measurement

The concentration of PGE_2_ in cell culture media was measured using the prostaglandin E_2_ ELISA kit (ab136948, abcam, Melbourne, Australia). The methods for sample preparation and analysis were performed according to the manufacturer’s instructions. The samples were analyzed using the microplate reader to measure the optical density set at 530 nm.

### 2.20. TNF-α Assay

The human TNF-α ELISA kit (ab181421, Abcam, Melbourne, Australia) was used to measure levels of the cytokine in cell culture media following the incubation of endothelial cells for 24 h with or without the PB, AMS-6PB or ascorbic acid. A microplate reader was used to measure the optical density of each sample at 450 nm as recommended by the supplier.

### 2.21. Flow Cytometry

The cellular uptake of FITC-AMS-6 was quantified by flow cytometry (CytoFLEX S, Beckman Coulter, Australia). Endothelial cells were grown in 24-well plates at an approximate seeding density of 150,000 cells per mL. When cells were approximately 90% confluent, FITC-AMS-6 in cell culture media was added to the cells and incubated for the experimental time points. After the incubation time, cells were washed with PBS, detached with Trypsin EDTA, and cell suspensions were centrifuged at 500 *rcf* for 5 min to remove the supernatant. The cell pellet was suspended in PBS for the fluorescent detection of FITC-AMS-6 by a flow cytometer. Ten thousand events were collected for each sample, and the results were analyzed using CytExpert software (Beckman Coulter).

### 2.22. Preparation of Endothelial Cells for Cellular Uptake of Probucol

Endothelial cells were seeded in 24-well plates at a density of 5 × 10^5^ cells/well. Confluent cells were treated with 1 μg/mL LPS together with either AMS-6PB30% 10 μM or PB 10 μM for 2, 4, 6 and 24 h. After the treatment time, media were removed and discarded, cells were washed three times with 500 μL PBS and cells were solubilized in 300 μL solution of 1% NP40 for 30 min. After 30 min, 700 μL of acetonitrile was added, the samples were centrifuged for 10 min at 10,000 rcf and were then filtered through a 0.45 um filter, and 500 μL was transferred to HPLC vials.

### 2.23. Preparation of Standard Samples

Probucol was dissolved in acetonitrile and diluted to known concentrations to prepare a standard curve between 100 μg/mL and 0.1 μg/mL. Eight calibration standards were prepared for probucol between 0.1 μg/mL and 100 μg/mL. The limit of quantitation was 0.1 μg/mL.

### 2.24. HPLC Conditions

The concentration of probucol was analyzed by HPLC (Agilent Technologies Inc., Santa Clara, CA, USA). The mobile phase was delivered at a flow rate of 1.0 mL/min through a C18 column (Zobrax SB-C18, 5 μm, 4.6 × 150 mm, Agilent Technologies Inc, Santa Clara, CA, USA) at 40 °C, and the detection wavelength was 241 nm. The injection volume was 1 μL. The mobile phase consisted of acetonitrile (solvent A) and distilled water (solvent B) delivered using a gradient: Solvent A was 80% at 0 min, 80% to 85% at 3 min and 85% to 90% at 7 min for a total run time of 10 min. An auto sampler injection volume of 5 μL was used for sample analysis.

### 2.25. Data Analysis

Analysis was carried out using Agilent Chem station software (V. A. 09. 01, Agilent Technologies Inc., Santa Clara, CA, USA). The concentrations of PB in the samples were determined by the linear equation obtained from the standard curve (*R*^2^ = 0.99).

### 2.26. TEER and FITC Dextran Permeability Measurements

For trans-endothelial electrical resistance (TEER) and permeability experiments, endothelial cells were seeded on pre-coated type I collagen Transwell^®^ filters (Corning^®^ Transwell^®^ Polyester membrane cell culture inserts, 24 well, 6.5 mm Transwell^®^ with 0.4 μm pore polyester membrane insert, Sigma, Australia) at a density of 50,000 cells/cm^2^. Cell culture media were changed every 2–3 days, and experiments were performed when cells were approximately ~90% confluent, 5–6 days after seeding. AMS-6PB or PB was added together with 1 μg/mL LPS at the start of the experiment, and TEER measurements were taken by using an EVOM2 voltmeter with STX-2 electrodes (World Precision Instruments) at 2-, 4-, 6- and 24-h time points. To calculate the TEER (Ω·cm^2^), electrical resistance across a collagen coated insert without cells was subtracted from the TEER readings obtained on inserts with cells, and this value was multiplied by the surface area of the insert.

### 2.27. FITC Dextran Permeability Coefficient Measurements and Calculations

The permeability of FITC dextran across the endothelial cell monolayer grown on Transwell^®^ inserts was determined after 24-h incubation of endothelial cells with 1 μg/mL LPS with or without AMS-6PB, ascorbic acid or PB. Two different sizes of FITC dextran were used for permeability measurements, a larger (59–77 kDa, FD70, Sigma, Australia) and a smaller (3–5 kDa, FD4, Sigma, Australia). A final concentration of 1 mg/mL FITC dextran in EBM2 cell culture media was applied to the apical donor compartment of the Transwell^®^ insert, and to detect the concentration of the FITC dextran accumulated in the well on the bottom chamber, 50 μL of media from the bottom well was taken at time points 0, 15, 30, 45, 60, 90 and 120 min and transferred to a black-walled 96-well plate. The fluorescence intensity of each sample was measured by a plate reader (FLUOstar OPTIMA, BMG LABTECH, Victoria) at an excitation wavelength of 480 nm and an emission wavelength of 520 nm. Concentrations were calculated using standard curves generated from the stock solution of FITC dextran.

### 2.28. Calculation of the Permeability Coefficient (P_e_)

Permeability coefficients take into account the difference in permeability of FITC dextran across the Transwell^®^ membrane with and without cells. The concentration of the FITC dextran added to the top (abluminal) of the Transwell^®^ membrane and collected from the bottom (luminal) well of the Transwell^®^ plate was calculated for each time point based on the standard calibration curve of FITC dextran. The cleared volume across the transwell^®^ membrane was calculated for each time point using the following equation [63]:$$Cleared\;volume\;\left(\mu l\right)=\frac{Concentration_{abluminal\;}\times Volume_{abluminal}}{Concentration_{luminal}}$$

The average cleared volumes were plotted versus time in minutes for each sample. Clearance slopes for the empty transwell^®^ membrane (PS_insert_) and the transwell^®^ membrane with endothelial cells following treatment with LPS and the test samples (PS_test +cells + insert_) were calculated using linear regression analysis and were used to obtain the permeability product of the test compound (PS_test+cells_) [63]:$$\frac{1}{PS_{cells}}=\frac{1}{PS_{cells+insert}}-\frac{1}{PS_{insert}}$$

Permeability coefficients (P_e_) for each compound were derived by dividing the PS_cells_ value by the surface area of the cell culture insert (0.33 cm^2^). Data are presented with units of × 10^−6^ cm/s.

### 2.29. Zebrafish Experiments

Zebrafish were maintained at 28 °C under a 13-h light and 11-h dark cycle. Zebrafish embryos were collected by natural spawning and were raised at 28 °C in buffer free-E3 solution (E3 solution) following the standard protocol [48,64].

### 2.30. Measurement of ROS Concentration in Zebrafish

The compound DCFDA (2′,7′-dichlorodihydrofluorescein diacetate) was used to monitor ROS production in embryonic zebrafish. DCFDA is cleaved by intracellular esterase and oxidised by ROS to the fluorescent compound DCF (2′,7′-dichlorofluorescein). Embryonic zebrafish (2dpf) were pre-treated with AMS-6PB or PB dissolved in E3 solution for 24 h in an incubator set to 28 °C. At 3 dpf, the embryos were washed three times in E3 solution and incubated with 5 mM H_2_O_2_ for one hour. The embryos were washed three times in E3 solution and treated with 25 mM DCFDA for 45 min, washed in E3 solution and transferred individually to a black 96-well plate. Fluorescence measurements were obtained using a PHERAstar plate reader (BMG LABTECH, Mornington, Victoria) set at excitation/emission wavelengths of 485/520 nm. Data are represented as the mean ± standard deviation. Samples were run in triplicate, and values were normalized to the positive control. Statistical significance was determined using one-way ANOVA.

### 2.31. Statistical Analysis

Unpaired Student’s *t*-tests were used for statistical analysis of in vitro and in vivo studies in zebrafish using Excel (Microsoft, Redmond, WA, USA). A *p*-value < 0.05 was considered statistically significant.

## 3. Results

### 3.1. Material Characterization

Mesoporous silica AMS-6 was synthesized following a protocol reported previously [32]. The structural and porous properties of the calcined MSP (CAL-AMS-6) were characterized by X-ray diffraction (XRD), scanning electron microscopy and N_2_ porosimetry (Figure 1a). Amine-functionalized MSP (NH_2_-AMS-6) and fluorescein-conjugated MSP (FITC-AMS-6) were also prepared, and detailed characterization is included in the Appendix A.

Similar to those observed for CAL-AMS-6, the XRD diffractograms of the NH_2_-AMS-6 material showed similar diffraction peaks that can be indexed on the basis of a cubic unit cell, suggesting that minimal changes in porous structure occurred (Appendix A) [65,66]. NH_2_-AMS-6 particles were then utilized to conjugate FITC within the pores, to visualize and measure the particle uptake in the in vitro and in vivo studies. A reduction in the intensity of the XRD diffraction peaks for the FITC-AMS-6 sample is likely due to a small loss of porous structure during the FITC conjugation procedure. Porosimetry measurements confirm that approximately 50% of the mesopore volume was filled in the FITC-AMS-6 sample (Appendix A). The presence of an FT–IR absorption band centered at 1462 cm^−1^ in FITC-AMS-6 was characteristic of the iminothioester bond between amino groups (-NH_2_) and the isothiocyanate groups (-N=C=S) from FITC [67]. The absorption band at 1630 cm-1 was characteristic of the -NH_2_ groups in NH_2_-AMS-6 (Appendix A) [67]. FITC-AMS-6 particles were subsequently used for uptake studies in endothelial cells and zebrafish.

### 3.2. Drug Loading and Dissolution Kinetics

Probucol (PB) was loaded into CAL-AMS-6 particles through a wetness impregnation method. The loading amount (wt.%) was determined from thermogravimetric analysis, TGA. A significant reduction in surface area and pore volume indicated that the drug was loaded within the mesopores (Table 1). The lack of a crystalline endotherm in the DSC traces or diffraction peaks in the high-angle XRD of loaded samples was evidence that the drug was loaded in an amorphous state (Figure 1b,c) [68]. As reported previously, drug loadings of PB higher than 30 wt.% led to the complete filling of the mesopore volume of AMS-6 and re-crystallization of the drug in the exterior of the particles [69]. To avoid this, samples with loading amounts of 29.5 wt.% AMS-6PB were prepared and used for cellular studies.

Crystalline PB has an extremely low aqueous solubility (~5 ng/mL) [70], and dissolution in simulated intestinal fluid (SIF) was extremely slow, with 20% of the compound solubilized in 20 h (Figure 1d). In contrast, PB released from AMS-6 in SIF achieved 100% drug release after 20 h, with 50% of the loaded drug released (T50%) in 1.6 h.

### 3.3. In Vitro Attenuation of Oxidative Stress

To assess the efficacy of our model of oxidative stress and inflammation in human brain endothelial cells (HBEC), the optimal LPS dose and incubation time required to generate intracellular ROS and cell death were first identified (Figure 2). The Muse oxidative stress assay was used to measure the percentage of cells that were positive for the generation of the primary ROS, the superoxide anion, O^2−^. Incubation for 24 h with 1 μg/mL LPS generated the highest percentage of ROS-positive cells (~25.4%, Figure 2a) compared to the negative control (*p* < 0.05), whilst longer incubation times did not enhance oxidative stress. Analysis of cell viability showed only a modest increase (approximate 3%) in cell death after incubation with 1 μg/mL LPS for 24 h (Figure 2b), which is an ideal experimental setting to study ROS production and inflammation.

Significant (*p* < 0.05) increases in the overall levels of ROS species (Figure 2c) were observed, including levels of hydrogen peroxide, hydroxyl, peroxyl, nitric oxide and peroxynitrite after treatment with 1 μg/mL LPS for 24 h. Inflammation markers such as the total COX enzyme activity, levels of PGE_2_ and cytokine TNF-α were also increased after incubation in 1 μg/mL LPS for 24 h compared to negative controls (*p* < 0.05) (Figure 2c). These results are consistent with previous in vitro reports on the generation of cellular oxidative stress and inflammation in endothelial cells treated with LPS [15], and together with the absence of pronounced cell death provided an optimal cellular scenario to study the ability of PB to quench ROS production and inhibit the COX-mediated production of inflammatory markers.

Crystalline PB and AMS-6PB were added to HBEC at different concentrations together with 1 μg/mL LPS and incubated for 24 h. For comparison, the water-soluble antioxidant ascorbic acid was also tested without encapsulation into a mesoporous particle carrier. Ascorbic acid undergoes rapid cellular uptake in endothelial cells, establishing a high intracellular content. Strikingly, the antioxidant properties AMS-6PB were comparable to ascorbic acid and were significantly enhanced when compared to crystalline PB, evident from a consistently lower number of ROS+ cells at all concentrations studied (1–100 μM). While crystalline PB itself had some effect in reducing the percentage of ROS-positive cells, AMS-6PB was significantly (*p* < 0.05) more effective (Figure 3a). Cell viability data correlated well with ROS inhibition in LPS-treated cells (Appendix A). The effects of our treatments on intracellular ROS, mitochondrial hydroxyl radicals, superoxide anions and peroxynitrite were then assessed (Figure 3b–e). AMS-6PB inhibited the production of the superoxide anion and peroxynitrite at comparable levels to ascorbic acid between 1 and 100 μM and that of mitochondrial hydroxyl radicals at comparable levels to ascorbic acid between 10 and 100 μM. Levels of nitric oxide were higher in cells treated with AMS-6PB and ascorbic acid at 10 and 50 μM compared to negative controls. Together, these data show a strong effect of PB release from AMS-6, which enhanced the antioxidative effects by ~40%, reducing the fraction of ROS-positive cells after LPS stimulation and having a comparable effect to ascorbic acid.

We also observed greater inhibition of total COX enzyme activity and lower levels of COX-dependent release of PGE_2_ for AMS-6PB compared with PB (Figure 4a,b). Since the COX enzyme resides intracellularly within the lumen of the endoplasmic reticulum and nuclear envelop [71], this provides strong evidence for the enhanced molecular transport of PB within the cell due to their faster release from the mesoporous carriers.

Surprisingly, levels of TNF-α remained above the LPS positive control for AMS-6PB (Appendix A) at all concentrations and for PB (except 1 μM). This effect was not associated with the AMS-6 silica particles, which overall decreased the concentration of TNF-α and PGE2 in cells after 24-h incubation with 1 μg/mL LPS (Appendix A) in a dose-independent manner. Cells treated with ascorbic acid showed decreased TNF-α levels. The cellular uptake of FITC-AMS-6 particles by endothelial cells was then measured by flow cytometry at different incubation times. Approximately 40% of endothelial cells were recorded as FITC positive after 6 h incubation of 50 and 100 μg/mL FTIC-AMS-6, which was not significantly different to the result after 24-h incubation, suggesting that a rapid uptake of mesoporous silica particles occurred (Figure 4c). This is consistent with published data on the uptake of AMS-6 particles by different cell lines [72,73]. The test compound AMS-6PB had a significantly higher concentration of intracellular PB after 24-h incubation than crystalline PB, as measured from HPLC analysis, with 46.2% of the loaded PB being transported intracellularly in contrast to 6.0% for PB alone (Figure 4d).

To test the effectiveness of the AMS-6PB in maintaining the integrity of a model BBB, we established an in vitro model consisting of HBEC cultured on the membrane of the transwell insert, prior to the addition of LPS (1 μg/mL) with or without AMS-6PB. Trans-endothelial electrical resistance measurements were performed using a TEER apparatus, which is correlated with BBB disruption associated with paracellular pathways [74]. For adequate comparison, the resistance data were normalized to the initial value measured before addition of the LPS and are expressed as a percentage. The initial resistance values varied for each transwell, between 450 and 510 Ω·cm^2^. In the negative control, which consisted of brain endothelial cells grown on the transwell membrane in cell culture media, a high TEER value was measured, which remained unchanged after 24 h (Figure 5a). This indicates a high degree of BBB integrity of the endothelial cell monolayer control. Treatment with LPS (positive control) resulted in a significant reduction in TEER in comparison to the negative control (Figure 5a). As noted, LPS is known to cause the generation of oxidative stress in endothelial cells that disrupts the integrity of the blood brain barrier through multiple mechanisms including the induction of cellular apoptosis, nitration and oxidation of tight junction proteins [8].

Mesoporous AMS-6 particles alone (without PB) added together with LPS led to a concentration-independent decrease in the resistance per unit area, despite the resistance observed for the positive control (LPS alone, Appendix A). AMS-6PB and ascorbic acid improved the HBEC monolayer integrity when added with LPS (after 24 h of incubation) in comparison to the positive control. Addition of AMS-6PB at 1 μM and 50 μM concentrations led to comparable decreases in resistance to ascorbic acid and a significant lower disruption of the monolayer at 100 μM (Figure 5a).

The enhanced dissolution properties of PB released from AMS-6 were revealed in the time-dependent change in integrity of (i.e., resistance) across the HBEC monolayer (Figure 5b–d and Appendix A). The highest resistance values were observed at 2 h for the 100 μM concentration, with an 18% improvement. Addition of either 50 μM of ascorbic acid or AMS-6PB led to the most sustained retention of the HBEC monolayer integrity over the 24-h period.

Membrane permeability to fluorescein isothiocyanate (FITC)-dextran molecules of molecular weight 59–77 kDa (approx. size 60 Å) and 3–5 kDa (approx. size 14 Å) was used as an indicator of the degree of disruption to tight junctions between HBEC after 24-h incubation with 1 μg/mL LPS and test compounds. Increased permeability to the larger size FITC-dextran represents broad disruption to the integrity of the endothelial cell monolayer while the smaller size FITC-dextran is a more sensitive marker of BBB disruption [75]. In comparison to TEER, permeability measurements are associated with disruption of the BBB by the transcytotic pathways [76]. As a reference, the positive control LPS (1 μg/mL) induced a 4-fold increase in the permeability rate of the larger size FITC-dextran compared to the negative control, and approx. 2.5 times higher permeability than the smaller FITC-dextran (Figure 6).

This indicates that LPS caused a broad disruption to the BBB integrity involving both paracellular and transcytotic pathways as measured by TEER and the permeability to tracer compounds [75,77,78]. This correlates with the increases in ROS and PGE_2_ levels observed in LPS-treated endothelial cells and with the broad disruption of the BBB associated with molecular mechanisms caused by opening of tight junctions and oxidative modification of the structure and function of biomolecules as mediated by oxidative stress and inflammation. Treatment with AMS-6PB and ascorbic acid maintained a lower permeability to the larger and smaller FITC-dextran compared to controls, confirming that these were effective in preventing cellular damage associated with LPS.

The antioxidant properties of PB released from AMS-6 as compared to crystalline PB and ascorbic acid were finally probed in an in vivo model of oxidative stress in zebrafish. Zebrafish were pre-treated with the test compounds for 24 h, followed by incubation with H_2_O_2_ at 5 mM for 1 h. The ROS probe, DCFDA (25 μM) is converted by intracellular ROS to the fluorescent product 2′-7′ dichlorofluorescein (DCF) and was used as a proxy to measure the intracellular oxidant concentration [52]. Fluorescence intensity of DCF was measured kinetically by a microplate reader every 10 min over 16 h, and results were normalized to the positive control (Figure 7a). Treatment with 5 mM H_2_O_2_ caused an approximate 2-fold increase in measured ROS concentration as compared to the negative control at both time points (0 and 16 h) (Figure 7a).

As expected, H_2_O_2_ treatments also resulted in a significant percentage of fish found dead at the final time point (*t* = 16 h) compared to the negative control (Figure 7b). This is likely due to the exposure to high concentrations of oxidative species that are known to modify cellular structure and function by mechanisms involving the oxidation of biomolecules including DNA, proteins and lipids and the promotion of cellular apoptosis and death during oxidative stress [68]. Strikingly, zebrafish pre-treated with AMS-6PB (25 µM and 50 µM) maintained comparable levels of ROS to the negative control (Figure 7a).

To confirm that the reduction in ROS was due to the released PB, zebrafish were pre-treated with CAL-AMS-6 alone at different concentrations for 24 h followed by 5 mM H_2_O_2_ (Appendix A). In this case, the measured ROS concentration and viability of zebrafish were not significantly different to those in the positive control (Appendix A), confirming that the release of PB from AMS-6 was responsible for the reduction in ROS concentration observed in zebrafish pre-treated with AMS-6PB samples at 25 and 50 μM concentrations (Figure 7a). Interestingly, we observed an increase in ROS at higher concentrations of AMS-6PB (100 and 200 μM) (Appendix A). This was likely triggered by the exposure of zebrafish to higher concentrations of the AMS-6 silica alone, as exposure of zebrafish to 100 and 200 μg/mL of CAL-AMS-6 also caused an increase in ROS and fish death after 24-h incubation time, without H_2_O_2_ treatment (Appendix A).

To examine this further, a dose-dependent examination of silica uptake and biodistribution in zebrafish was conducted by fluorescent microscopy (Figure 8). Most of the FITC-AMS-6 particles were in and around the frontal regions of the fish including the heart, yolk sac and mouth, and greater accumulation of FITC-AMS-6 in these regions was observed at higher concentrations of silica (100 and 200 μg/mL). It is likely that the accumulation of AMS-6 particles at higher doses was the cause of the increase in ROS production as observed in Appendix A and consequently contributed to the production of oxidative stress resulting in increased fish death. This is also consistent with previous findings, where zebrafish exposed to high concentrations of MSPs (>100 μg/mL) resulted in higher penetration and biodistribution of the particles, increased ROS production, reduction in antioxidant enzyme activity and altered gene expression [79,80]. Notably, PB mitigated the effects of mesoporous silica. The reduction in ROS was significantly greater for AMS-6PB than for crystalline PB (Figure 7a). Pre-treatment at high concentrations of ascorbic acid (200 µM) was required to maintain levels of ROS at comparable levels to the negative control and to AMS-6PB 25 and 50 μM. After 16 h, the measured ROS levels in zebrafish were not significantly different to that observed at the measurement taken at time point 0 (Figure 7a and Appendix A). This was also reflected with a high percentage of fish that remained alive after treatment with AMS-6PB and ascorbic acid, with ~91% of fish alive recorded in the AMS-6PB30% 25 µM treatment (Figure 7b,c and Appendix A).

## 4. Discussion

Overall, our results point towards an enhancement in the antioxidant properties of PB when released from mesoporous silica particles in comparison to the corresponding non-encapsulated compounds. A decrease in the percentage of ROS+ cells was observed. The test compound AMS-6PB was efficient at scavenging ROS, acting to reduce levels of superoxide anion, peroxynitrite and mitochondrial hydroxyl radicals to a similar degree as ascorbic acid. These are highly reactive ROS that can cause oxidative damage to the structural integrity of the BBB [81,82]. In addition to their ROS-scavenging properties, PB and ascorbic acid are known to preserve the enzymatic activity of endothelial nitric oxide synthase (eNOS) that produces nitric oxide at physiological levels to maintain the structure and function of the BBB [6,83,84,85,86,87]. Nitric oxide produced by the eNOS enzyme reduces endothelial permeability via the nitric oxide-guanosine 3′,5′- cyclic monophosphate (cGMP) mechanism [24,81,82,88,89,90].

Probucol has anti-inflammatory properties that are mediated by the inhibition of NF-κB-mediated activation of COX enzyme activity [19]. Levels of PGE_2_ were lower for AMS-6PB treated cells than for ascorbic acid. Thus, PB released from AMS-6 was as effective in the reduction of LPS-induced increases in COX-mediated PGE_2_ production. Treatment of brain endothelial cells with 1 μg/mL LPS after 24 h generated a significant but small increase in the measured TNF-α concentration of 90 pg/mL as compared to 68 pg/mL in the negative control (Appendix A). This finding is consistent with previous reports that brain endothelial cells produce less than 100 pg/mL of TNF-α when challenged by LPS [91]. Surprisingly, suppression of ROS and decreased levels of PGE_2_ did not result in a reduction in the levels of TNF-α when cells were treated with AMS-6PB, except at 1 μM concentration, and were otherwise higher than those in the negative control. This is not associated with the mesoporous silica particles themselves which did not induce higher levels of TNF-α in comparison to the negative control (Appendix A). However, AMS-6 particles alone did induce significant higher levels of PGE_2_.

Doses of TNF-α between 1 and 100 ng/mL have been previously shown to induce a significant increase in the permeability of an in vitro BBB model consisting of a monolayer of cultured brain endothelial cells [77,92]. In those previous studies, the increase in BBB permeability was dependent on the dose of TNF-α and the incubation time, with a significant decrease in BBB permeability measured at short incubation times between 2 to 6 h of the cytokine. In this work, no decreases in permeability were observed during that time interval except for the positive control. The small increases in COX-mediated PGE2 did not affect the endothelial cell monolayer integrity in AMS-6PB-treated cells, as only small changes in resistance were observed. Thus, treatment with AMS-6PB was more effective in mitigating LPS-mediated disruption to the BBB specially at shorter time intervals, through mechanisms involving quenching of reactive species that protects against oxidative modification of biomolecules and preservation of nitric oxide function that sustain endothelial cell structure and function in maintaining the integrity of the BBB. The higher solubility profile of PB originating from its encapsulation in an amorphous form within the mesopores likely facilitated the greater antioxidant activity, neutralizing both intracellular and extracellular H2O2 present in biological media. Furthermore, the release of PB from AMS-6 may have facilitated its uptake and incorporation into the biological membranes protecting against H2O2-mediated oxidative damage and activation of cellular apoptosis [93,94,95]. Other potential beneficial effects of PB that may play a role in the suppression of BBB disruption include an increase in antioxidant enzyme glutathione peroxidase (GPx) activity that catalyzes the reduction of H2O2 into unreactive products [96] and increased protein expression of thioredoxin- 1 [93] and NAD(P)H:quinone oxidoreductase-1 [97], all of which are mechanisms that can contribute to attenuation of oxidative stress caused by H_2_O_2_. Indeed, from zebrafish experiments, PB released from AMS-6 (at 25 and 50 µM) was comparable in mitigating the effect of oxidative stress caused by H_2_O_2_ as ascorbic acid at a 200 µM dosage. As a potent water-soluble antioxidant, ascorbic acid can neutralize a range of ROS and RNS present in biological media and can restore the antioxidant activity of other compounds and enzymes in vivo [98,99]. Exposure of freshly fertilized zebrafish embryos to ascorbic acid at 100–200 µM has been shown to promote embryo development and fish growth and to reduce ROS DNA damage and the hatching rate [100]. A high concentration of ascorbic acid is likely required to achieve and maintain high levels of intracellular ascorbate levels required to quench ROS during oxidative stress.

Whilst oral administration of mesoporous silica particles is a viable formulation route to increase the bioavailability of probucol, [101,102] delivery of PB to the central nervous system (CNS), associated with oxidative stress, may lead to neuro-inflammation, facilitated by the design of a nasal delivery vehicle with mesoporous excipients [103].

## 5. Conclusions

The enhancement in the solubility of PB released from AMS-6 increased its antioxidant properties in comparison to the administration of the crystalline compound in an in vitro endothelial cell model of the BBB. Reduction in ROS was higher for PB released from AMS-6 than PB with inhibition of mitochondrial hydroxyl radical generation and COX-meditated PGE_2_. The antioxidant properties of PB released from AMS-6 were comparable to those of the potent antioxidant ascorbic acid, maintaining a high human endothelial cell monolayer and tight junction integrity under oxidative stress and decreasing the permeability across the endothelial monolayer. AMS-6PB mitigated the effects of oxidative stress in zebrafish at lower concentrations than ascorbic acid. However, PB antioxidant properties in zebrafish were likely affected by an increase in agglomerated silica particles and their accumulation in certain organs when administered at high concentrations, which led to higher levels of ROS and a decrease in the viability of the fish. Levels of TNF-α in AMS-6PB-treated endothelial cells were not reduced in comparison to the LPS control except at the lowest concentration. This measurement was not due to the silica particles and should be investigated further.

Overall, this study continues to demonstrate the potent antioxidant properties of probucol and its potential effective use in the treatment of neuroinflammation. Further studies of probucol released from mesoporous silica particles in the context of hypocholesteremia and atherosclerosis, where it was shown to be therapeutically relevant, [87,104,105,106] may lead to new mechanistic insights.

## Figures and Tables

**Figure 1 pharmaceutics-14-00502-f001:**
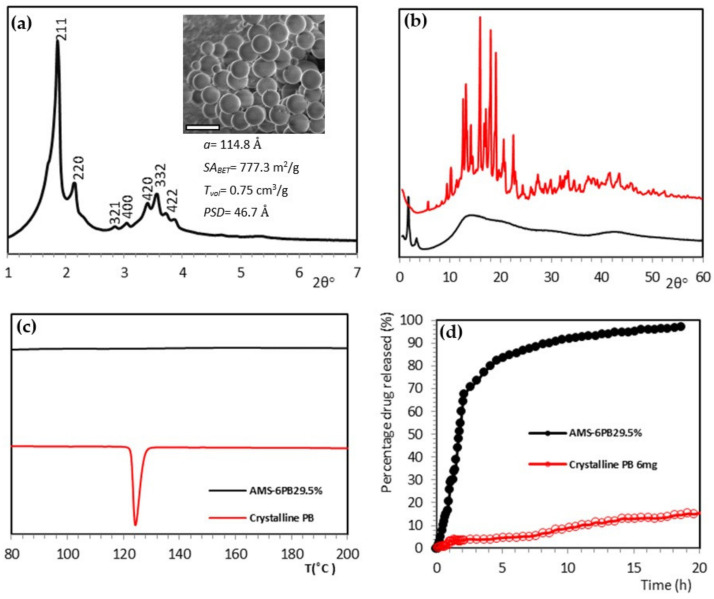
(**a**) XRD showing mesoscale peaks for bicontinuous cubic AMS-6. Inset shows a representative SEM image of AMS-6 particles (scale bar = 1 μm). *a*= unit cell parameters, *SA_BET_* = surface area, *T_vol_* = total pore volume, *PSD* = pore size distribution, *PS* = average particle size; (**b**) XRD of crystalline PB and AMS-6PB. The *y*-axis in (Figure 1a,b) represents intensity in arbitrary units; (**c**) DSC analysis of crystalline probucol (PB) compared to PB loaded in AMS-6; (**d**) Percentage (%) drug released for crystalline PB and AMS-6PB at an equivalent dose. Measurements conducted under sink conditions in SIF as the dissolution media (pH 6.8, 37 °C).

**Figure 2 pharmaceutics-14-00502-f002:**
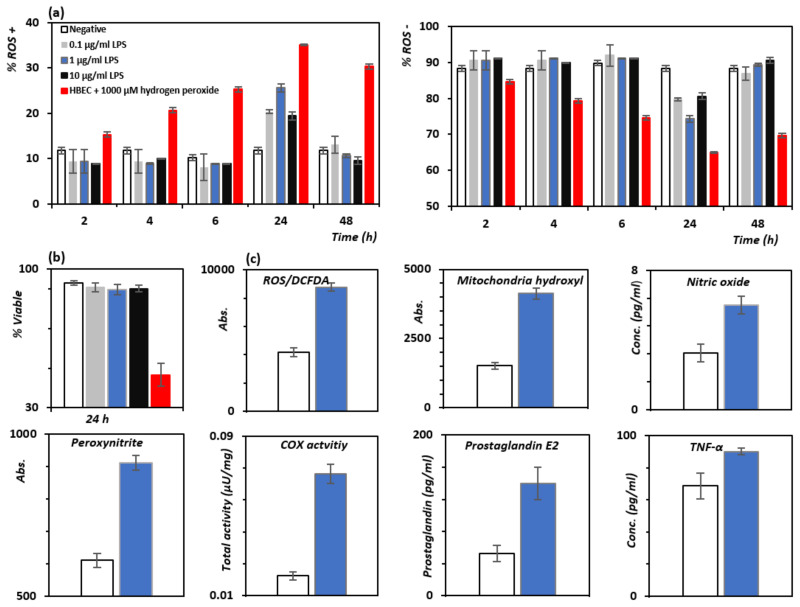
Percentage of (**a**) ROS+ and ROS- human brain endothelial cells (HBEC) after incubation with LPS at different test concentrations compared to the negative control (cells with media only) and the positive control (1000 μM hydrogen peroxide). (**b**) Cell viability assay. Production of ROS only caused a relatively small amount of cell death in comparison to the negative control. (**c**) Analysis of ROS levels using the DCFDA assay suggested an increase in hydrogen peroxide, hydroxyl and peroxyl levels, consistent with increases in mitochondrial hydroxyls, nitric oxide and peroxynitrite after incubation of 1 μg/mL LPS for 24 h in HBEC cells compared to the negative control (clear bar). Markers of inflammation: COX activity, prostaglandin E2 and TNF-α levels are also shown. Results show the mean ± SD of triplicates. Treatment with LPS resulted in significant increases in ROS and inflammatory markers (*p* < 0.05) as compared to negative controls.

**Figure 3 pharmaceutics-14-00502-f003:**
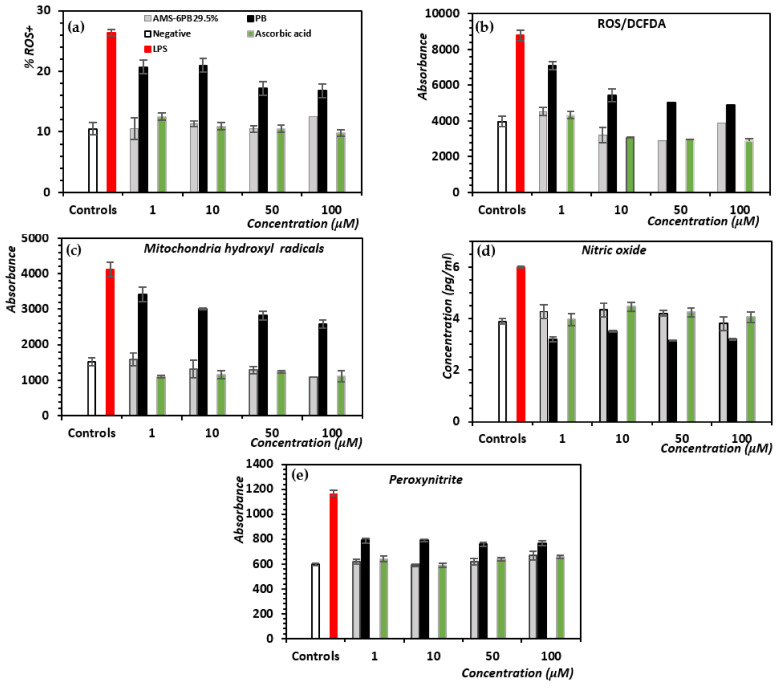
Concentration-dependent effects of ascorbic acid, AMS-6PB or PB (1–100 μM) on the production of different ROS by HBEC cells after 24-h incubation with 1 μg/mL LPS. (**a**) %ROS+ cells; (**b**) analysis of intracellular ROS (hydrogen peroxide, hydroxyl and peroxyl ions) using the DCFDA; (**c**) mitochondrial hydroxyl levels; (**d**) nitric oxide, and (**e**) peroxynitrite. The positive control (red bar) was cells incubated with 1 μg/mL LPS, and the negative control (white bar) was cells incubated with media only. Results show the mean ± SD of triplicates.

**Figure 4 pharmaceutics-14-00502-f004:**
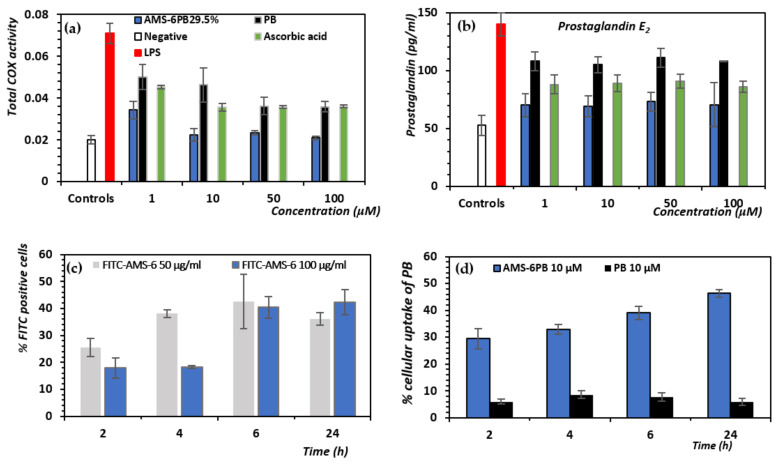
Concentration-dependent effects of AMS-PB and PB on (**a**) the total COX activity; (**b**) prostaglandin E2 levels. (**c**) Uptake studies of FITC-AMS-6 analyzed by flow cytometry following incubation at different time points of FITC-AMS-6 at 50 and 100 μg/mL concentration in endothelial cells. (**d**) Percentage uptake of PB in endothelial cells as measured by HPLC on HBEC after 24-h incubation with 1 μg/mL LPS. The positive control (red bar) was cells incubated with 1 μg/mL LPS, and the negative control (white bar) was cells incubated with media only for 24 h. Results show the mean ± SD of triplicates.

**Figure 5 pharmaceutics-14-00502-f005:**
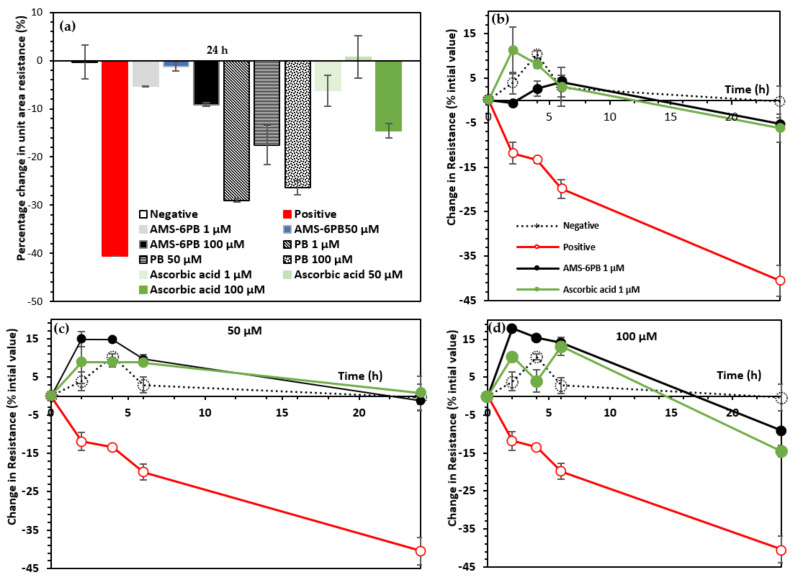
(**a**) TEER electrical resistance measurements conducted on monolayers of HBEC, as a BBB integrity model, incubated with 1 μg/mL LPS with PB, AMS-6PB or ascorbic acid after 24-h incubation, expressed as a percentage decrease from the initial measured resistance prior to the addition of LPS. Time-dependent resistance measurements at (**b**) 1 μM (**c**) 50 μM and (**d**) 100 μM for AMS-6PB and ascorbic acid compared to controls. For clarity, the error bars for the negative and positive controls were removed in (**c**,**d**). For comparison, all data were normalized to the value at 0 h prior to addition of the LPS. The final resistance values for the positive and negative control were 290 and 450 Ω·cm^2^, respectively.

**Figure 6 pharmaceutics-14-00502-f006:**
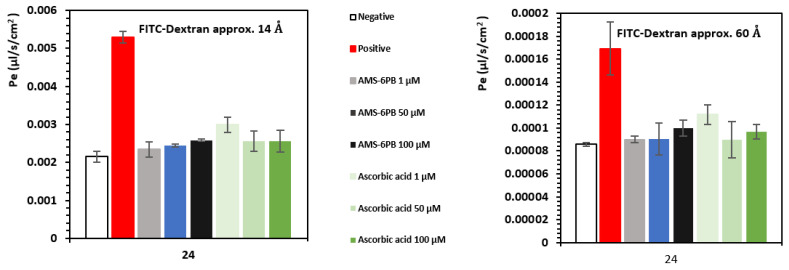
HBEC permeability (Pe) measurements to FITC–dextran molecules of different molecular weight and size, recorded at the end of the 24-h incubation period with 1 μg/mL LPS and test compounds.

**Figure 7 pharmaceutics-14-00502-f007:**
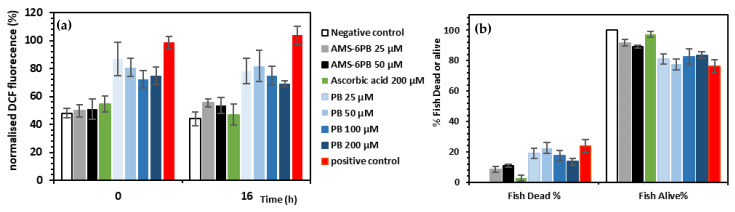
Embryonic zebrafish (2dpf) pre-treated with test compounds for 24 h, followed by 1-h treatment with 5 mM H_2_O_2_ and 45-min incubation with 25 μM DCFDA ROS probe. (**a**) Fluorescent intensity of DCF normalized to the positive control (5 mM H_2_O_2_, expressed as % over 16 h); (**b**) Percentage of fish dead or alive after 16 h as observed under brightfield microscopy (*n* = 6).

**Figure 8 pharmaceutics-14-00502-f008:**
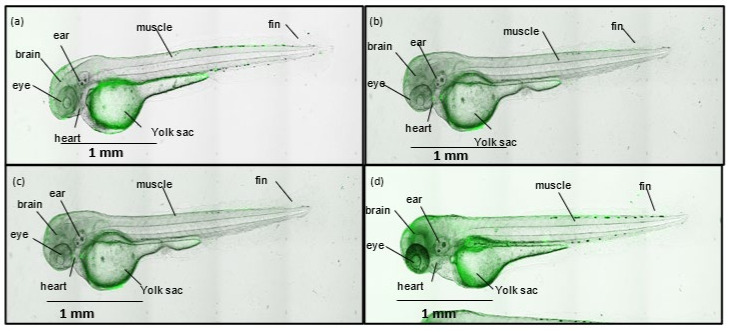
Brightfield images of zebrafish after 24 h exposure to FITC-AMS-6, at concentrations of (**a**) 25 µg/mL, (**b**) 50 µg/mL, (**c**) 100 µg/mL, and (**d**) 200 µg/mL.

**Table 1 pharmaceutics-14-00502-t001:** Physical properties of free probucol (PB) and drug-loaded AMS-6 particles.

Material	*a*^a^ (Å)	SA_BET_ ^b^ (m^2^/g)	T_vol_ (cm^3^/g)	PSD (Å)	D_T_ ^c^ (°C)	T_m_ ^d^ (°C)	ΔH_m_ (J/g)
CAL-AMS-6	114.8	777.3	0.75	46.7	-	-	-
PB	-	-	-	-	210.4	124.2	54.5
AMS-6 PB (29.5%)	113.9	197.7	0.21	40.3	256.5	-	-

^a^ XRD unit cell parameter (*a*), ^b^ BET surface area (SA_BET_), total pore volume (T_vol)_ and pore size distribution (PSD) calculated from N_2_ adsorption isotherms, ^c^ drug decomposition temperature (D_T_) from TGA, ^d^ Melting endotherms (T_m_) and enthalpy (ΔH_m_) from DSC.

## Data Availability

Data are contained within the article and Appendix A.

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
