# Peer review of "Enhanced Antioxidant Effects of the Anti-Inflammatory Compound Probucol When Released from Mesoporous Silica Particles"

_pharmaceutics, 2022, doi:10.3390/pharmaceutics14030502_

Round 1

Reviewer 1 Report

The manuscript by Lau et al on the role of formulation of mesoporous particles of Probucol and indomethacin in the enhanced antioxidant effects of these compounds is novel, data-driven and well-written. Few minor suggestions are provided below to improve the manuscript prior to its final publication.

  • Since probucol shows significantly enhanced invitro as well as invivo performance as compared to indomethacin data in the manuscript. The authors are suggested to focus the manuscript on probucol data and remove the indomethacin data if possible for succinct narrative and greater impact of the publication. The manuscript is otherwise very lengthy.
  • Use term test formulation for AMS-6PB and AMS-6INDO rather than test compound to differentiate the formulation from the compounds themselves.
  • Figure 4 is significantly distorted in the pdf version of the manuscript at the moment, authors are advised to either paste it in the picture format or ensure the translatability of the content in its original form when converted in the pdf version.
  • Authors should atleast provide commentary in the discussion around the translation of their invitro dissolution results with cellular assay results. In other words, how do authors envision a mesoporous silica based oral formulation translating into enhanced invivo performance in preserving the integrity of BBB in presence of oxidative stress?

Author Response

    • Since probucol shows significantly enhanced invitro as well as invivo performance as compared to indomethacin data in the manuscript. The authors are suggested to focus the manuscript on probucol data and remove the indomethacin data if possible for succinct narrative and greater impact of the publication. The manuscript is otherwise very lengthy.

    We thank the reviewer for the comment. We have done as suggested and removed all references to AMS-6INDO, focusing on the main results for AMS6PB and its comparison with PB alone and ascorbic acid. Indeed, we do believe that the manuscript is less "dense" and the most impactful result is now easier to find. A green highlight has been inserted wherever a change (a section taken out or the phrase revised so take out the discussion around INDO) has occurred. 

    Additionally, to make the manuscript less dense and due to the removal of the INDO data, we have merged Figures 1-3. All the figure numbers have been adjusted to reflect the new order. 

    • Use term test formulation for AMS-6PB and AMS-6INDO rather than test compound to differentiate the formulation from the compounds themselves.

    This has been corrected throughout the text. 

    • Figure 4 is significantly distorted in the pdf version of the manuscript at the moment, authors are advised to either paste it in the picture format or ensure the translatability of the content in its original form when converted in the pdf version.

    We have improved the resolution of the Figure and ensured that this is shown correctly in the pdf version of the text. 

    • Authors should at least provide commentary in the discussion around the translation of their invitro dissolution results with cellular assay results. In other words, how do authors envision a mesoporous silica based oral formulation translating into enhanced invivo performance in preserving the integrity of BBB in presence of oxidative stress?

    This is an interesting question. Whilst we anticipate that oral administration is the easiest route to deliver probucol in the context of enhancing its bioavailability, we also acknowledge that this may not be the most efficient route for rapid and efficient uptake in the context of the BBB. Our translation efforts instead will focus on the development of nasal delivery system, combining our AMS-6PB particles with mucoadhesive compounds that can efficiently deliver PB to the brain via the nasal route. We have written a sentence to this effect at the end of the discussion. We thank the reviewer for pointing this out. 

Reviewer 2 Report

The manuscript presents a systematic work on the antioxidant effects of two anti-inflammatory drugs loaded into AMS-6. The authors demonstrate an enhanced antioxidative effect of indomethacin and probucol assisted by an improved solubility of both compounds trough the encapsulation into the silica mesopores. The synergistic effect of these properties have the potential to preserve the structure and function of the BBB.
The paper is a very interesting study that could be oof a high interest for the reader, but the novelty of the work is not enough emphasized. Also, because of the complexity of the work, a schematic presentation of the idea can be useful. There are no major mistakes, but some sentences need to be revised for correction of typos. Some figures must be redraw: i) different colors are preferred instead gray shades; ii) SEM scale bare is not visible; iii) in fig. 4 red is missing from the legend and the name of the analyzed system is difficult to read.

Author Response

There are no major mistakes, but some sentences need to be revised for correction of typos. Some figures must be redraw:

i) different colors are preferred instead gray shades;

ii) SEM scale bare is not visible;

iii) in fig. 4 red is missing from the legend and the name of the analyzed system is difficult to read.

We thank the reviewer for these corrections which have now been implemented. Where ever possible the colors of the figures has been adjusted to remove greyscale contrast. 
